# Key Factors That Contribute to the Development of Resilience in Successful Women Leaders Who Experience Disrespect and the Importance of Respect in the Post-Pandemic Workplace

Carrie Spell-Hansson

Institute for Social Innovation, Fielding Graduate University, Santa Barbara, CA 93105, USA;
carrie@thefolkeinstitute.com

**Abstract:** Extrinsic structural inequities, such as historical biases against women in certain professions, their delegation to lower-paying jobs, gender, racial, and other discrimination, and additional systemic factors have been extensively studied as barriers to women entering and advancing in leadership positions in the workplace. Yet, the intrinsic individual characteristics of successful women leaders, including self-awareness, self-respect, self-esteem, self-confidence, self-acceptance, and resilience, that have facilitated their success in obtaining and retaining leadership positions despite these barriers have received far less attention in the literature. Resilience, in particular, is an important intrinsic characteristic that facilitates women's ability to navigate the often-difficult terrain of organizations, including facing disrespect by supervisors and colleagues. This study investigated the critical factors that contributed to the development of resilience among 24 successful women leaders in the United States which allowed them to be effective when experiencing disrespect in the workplace. Participants identified four categories of disrespect commonly experienced in the workplace, including: (1) not being listened to; (2) not being respected; (3) not being acknowledged; and (4) condescension. Factors that helped them develop the resilience to succeed despite these experiences included early developmental influences, circumstances they successfully overcame in life, and experiences in their youth that shaped how they responded as adults to disrespect or a lack of respect from their supervisors and colleagues. Participants also highlighted the importance of respect, the flip side of disrespect, in motivating them and enhancing their engagement in their work. The reported study is significant in that it identified factors that can be inculcated in women to help them develop resilience, and it highlighted the critical importance of creating a post-pandemic workplace that fosters mutual respect and does not tolerate disrespect.

**Keywords:** respect; disrespect; resilience; women leaders; COVID-19 pandemic

## 1. Introduction

Extrinsic structural inequities, such as historical biases against women in certain professions, their delegation to lower-paying jobs, gender, racial, and other discrimination, and additional systemic factors, have been extensively studied as barriers to women entering and advancing in leadership positions in the workplace. Yet, the intrinsic, individual characteristics of successful women leaders, including self-awareness, self-respect, self-esteem, self-confidence, self-acceptance, and resilience, that have facilitated their success in obtaining and retaining leadership positions despite these barriers have received far less attention in the literature. Resilience, in particular, is an important intrinsic characteristic that facilitates women's ability to navigate the often-difficult terrain of organizations, including facing disrespect by supervisors and colleagues [1]. The negative impact of disrespect and its flip side, respect, have been highlighted as important interpersonal factors that either work against or nurture employee commitment and job satisfaction and contribute to either a toxic or healthy work environment [1–6].

*1.1. Resilience*

Resilience has become an increasingly important ability to enable employees to survive and succeed in the turbulent and volatile work world characterized by complexity, rapidly changing local and global conditions, rapidly changing job configurations and downsizing, and external shocks such as the pandemic which displaced millions of workers and forced millions of others to work from home. Resilience is also an important ability for employees to overcome negative interpersonal interactions at work, especially those that derive from gender, racial, and other forms of discrimination. While most researchers agree that resilience is the ability to grow and move forward in the face of misfortune, there is still ambiguity surrounding the underlying process that comprises resilience [1].

For purposes of this study, resilience is defined as the individual's ability to adjust to adversity, maintain equilibrium, and retain or regain some sense of control over their environment and continue to move positively [1,2]. Luthar et al. [3] and Tugade and Fredrickson [4] defined adversity as the state of hardship or suffering associated with misfortune, trauma, distress, difficulty, or a tragic event. Workplace adversity can be defined as any negative, stressful, traumatic, or difficult situation or an episode of hardship encountered at work that creates barriers to role success or thriving in the organization [1].

Discussions on resilience being innate or learned are ongoing among researchers [1], including whether or not it requires positive growth or successful adaptation [2]. Resilience is not static. It is an active process, a balance between vulnerability and elasticity [2–4]. If equilibrium is maintained, an individual can theoretically manage any situation that comes along. Developing personal resilience can reduce vulnerability [5]. Individuals can develop and strengthen personal resilience by developing strategies for reducing their vulnerability and the personal impact of adversity in the workplace. Everyone has the potential to be resilient. One's resilience level is determined by factors such as individual experiences, the environment, and a balance of risk and protective factors [4].

London [6] asserted that the individual characteristics related to career motivation and success include career identity, career insight, and career resilience—"a person's resistance to career disruption in a less than optimal environment" (p. 621). Career resilience includes the ability to satisfactorily handle poor working conditions while one is aware of them. This ability includes self-efficacy, self-esteem, adaptability, and internal control, as well as risk-taking, low fear of failure, and a high tolerance for uncertainty [6].

Pincott [7] conducted in-depth interviews of 20 executive women leaders in nine industries in the United States to understand their conceptions of and strategies for developing and applying resilience. She found that these leaders conceived of resilience as the ability to bounce back, self-awareness, mind and body wellness, an optimistic outlook, adaptability, and the determination to succeed. Interviewed leaders identified the manifestations of resilience in the workplace as strategic thinking, social awareness, relationship management, building influential networks, credibility, and courage.

Ijames [8] gathered stories of resilience from ten African American women school principals in North Carolina who described how they exercised resilience in the face of the diminishment they experienced from gender and racial stereotypes and the challenges of leadership in general. They attributed their resilience to being armored by faith, family, community, and culture; being armored yet vulnerable; being undeterred and self-agentic; and fighting the good fight for purposeful leadership.

The six African American women superintendents of schools in the United States interviewed by Johnson [9] credited a number of factors to account for their resiliency in the face of enormous challenges and even adversity typical in their positions. They cited being raised in supportive families and communities, having supportive parents and key mentors in their early adulthood and early careers, being strengthened by faith and optimism, nurturing a healthy mind, body, and work–life balance, and treating everyone with respect and integrity.

Research indicates that while women workers are more likely to experience burnout than their male counterparts [10], women leaders have demonstrated more resilience

despite additional stress and exhaustion. Many have become stronger leaders who take on additional work associated with the new work environment [10].

*1.2. Disrespect*

Disrespect is experienced as a type of adversity in the workplace. Data resulting from a poll of 800 participants, managers, and employees revealed that 80% of participants lost work time worrying about a disrespectful incident and 78% said that their commitment to the organization declined [11]. Further, 48% of employees surveyed claimed they were treated uncivilly at work at least once a week; three out of four employees were dissatisfied with the way their company handled incivility [11]. Examples of how employees defined workplace disrespect included: (1) taking credit for others' efforts, (2) passing blame for one's own mistakes, (3) talking down to others, (4) not listening, (5) spreading rumors about colleagues, (6) making demeaning or derogatory remarks to someone, (7) withholding information, (8) belittling others' efforts, and (9) not saying please or thank you.

The 2019 research study entitled *Women in the Workplace* conducted by LeanIn.Org and McKinsey & Company [12] indicated that women experienced disrespect more often than their male counterparts in the workplace. The percentage was even higher for Black women, women with disabilities, lesbians, and bisexual women, as shown in Table 1. This systemic inequity plays out in workplaces all the way up to women leaders in the C-Suite [12]. This perceived lack of respect is an underlying cause of worker disengagement and discontent [12] which impedes the efforts of individuals and organizations to be resilient in adapting to the post-pandemic world.

**Table 1.** Disrespect Experienced by Women versus Men.

| | All Men | All Women | Lesbian Women | Bisexual Women | Women with Disabilities | White Women | Asian Women | Latinas | Black Women |
|---|---|---|---|---|---|---|---|---|---|
| Being mistaken for someone at a much lower level | 9% | 18% | 15% | 27% | 21% | 17% | 18% | 16% | 20% |
| Hearing demeaning remarks about you or people like you | 11% | 16% | 24% | 25% | 27% | 15% | 12% | 16% | 18% |
| Hearing others' surprise at your language skills or other abilities | 8% | 14% | 16% | 24% | 21% | 11% | 16% | 18% | 26% |
| Feeling like you can't talk about yourself or your life outside work | 7% | 10% | 23% | 26% | 21% | 10% | 8% | 9% | 12% |

Reprinted with permission from LeanIn.Org. [12]. Women in the Workplace. https://womenintheworkplace.com/2019 (accessed on 9 September 2022).

Disrespect occurs when an individual perceives that another does not acknowledge and show appreciation and value for their contribution and presence. Such disrespect, coupled with increased pandemic-related pressures, has caused approximately 2 million women to consider leaving the workforce completely or taking a step back from their careers [13]. Women are 1.3 times more likely than men to consider leaving the workforce, particularly senior women, Black women, and mothers [13]. These real and potential losses represent over 100,000 women in senior leadership positions. Unfortunately, Burns et al. [10] found that because the critical work women working are doing is not respected, their work is unacknowledged and unrewarded, many organizations risk losing capable women in their leadership ranks. The *Women in the Workplace* research indicated that only 32% of women and 50% of men believe disrespectful behavior toward women is often quickly addressed in their organization [12].

*1.3. Respect*

While disrespect has a negative impact on women leaders in the workplace, leaders and workers, in general, are more engaged, happier, and more likely to remain with an

organization when they feel respected [12]. Respect is often conceived of as "the state of being treated politely or being properly recognized for behavior" [14]. A culture of respect is essential for an organization to thrive and employees who feel respected feel valued and "thus often invested in developing their professional identity within their organization and cooperating with their teams, thus fostering organizational commitment" [14]. Individual respect translates into an enhanced collaborative process that strengthens not only personal identity and self-efficacy, but also the identity and efficacy of everyone in the group, team, or organization.

Huo and Binning [15] implied that there are two aspects to respect: one that supports the functioning of the collective (the organization) and another that supports the well-being of the individual (personal). Receiving respect at work supports the individual's ability to develop or strengthen positive self-identity that can result in positive personal and work-related outcomes [16]. Organizations that make creating a respectful environment a priority have found that respect contributes to job satisfaction and employee engagement [15]. Respect is one behavior that could lead to greater employee engagement and commitment [17].

Being treated with respect was the top contributor to overall job satisfaction based on a survey conducted by the Society of Human Resources Management [18]. Seventy-two percent of employees at all levels rated being treated with respect as very important. Organizations that make creating a respectful environment a priority have found that respect contributes to job satisfaction and employee engagement [19]. Showing respect can lead to greater employee engagement and commitment [19]. Furthermore, respect contributes to an inclusive environment. When managers and leaders respect all employees, they will treat all employees with equal value, build relationships with them, and create an environment free of discrimination and harassment. Research indicates that being valued and treated with respect can help create a positive work environment where employees feel fulfilled, loyal, engaged, and motivated to perform at their very best [16]. According to a survey of 20,000 employees conducted by Porath [20] for the Harvard Business Review, respect was the leading behavior that encouraged greater commitment and engagement. Pearson and Porath found that respectful behavior in the workplace was declining [11]. On the contrary, as Pearson and Porath [11] found, disrespect has increased in organizations. What was lacking in their analysis concerns why.

LaGree et al. [14] found that respectful communication had a positive impact on building resiliency, engagement, and job satisfaction by surveying 1,036 young workers in the United States from ages 21–34. The authors divided respect into respectful engagement and autonomous respect. They defined respectful engagement as occurring through "the relationship among team members, being interpersonally accepted, valued, and affirmed as part of a team" and autonomous respect as being personally accepted and respected by the organization in a way that coincides with "an individual's internal standards," similar to particularized respect as defined by Rogers et al. [21].

In other words, autonomous respect would manifest when organizational members would communicate with individuals in a manner that matched the individual's values of respectful communication, such as, for example, commending an individual for their ideas, contributions, and achievements. Autonomous respect would also manifest when members communicate with team members with words, such as "please" and "thank-you," using last names in greeting until requested to use first names, and other communication styles consistent with individual team members internal values of respect. They concluded that autonomous respect was an even stronger generator of resilience than respectful engagement. Their study highlights the importance of respect not only to strengthen engagement and job satisfaction but also to help build employees' ability to effectively navigate turbulence and external shocks such as the pandemic, as well as incidences of disrespect and other adversity in the workplace.

*1.4. Purpose of the Study*

The importance of resilience, the identification of disrespect as an impediment to employee engagement, and the important role of respect in fostering job satisfaction have been illustrated in the above literature review. Given these findings, this study examined the question, What factors impact and account for the resilience of women leaders who experience disrespect in the workplace?

The study was conducted to contribute to the literature regarding how women leaders can develop resilience. It provides information to enable coaching for women so that they are aware of the type of disrespect they may experience and how they can develop resilience in the face of such disrespect. The study also provides convincing evidence for the types and importance of respect in the workplace and prioritizes respectful workplaces for women in the post-pandemic world.

## 2. Materials and Methods

This study employed the qualitative methodology of narrative inquiry and thematic analysis. Narrative analysis has been used in many disciplines to learn more about the narrator's culture, historical experiences, identity, and lifestyle [22]. There are many analytic methods or forms of narrative analysis, including inquiry directed at narratives of the human experience or inquiry that produces data in narrative form [22]. Research is a mutually constructed story, a collaborative effort between the researcher and the participant, and narrative inquiry is a view of the phenomena of people's experiences [23]. It is a methodology that allows for the intimate study of the subject's experiences over time and in context.

Narrative thematic analysis in which the content within the text is the primary focus is the most common approach employed in narrative analysis and is the approach employed in this study. This approach generally follows five stages, and these stages were followed in this study: (1) organization and preparation of the data, (2) obtaining a general sense of the information, (3) the coding process, (4) categories or themes, and (5) interpretation of the data [22]. Thematic analysis is often selected because "it offers a toolkit for researchers who want to do robust and even sophisticated analysis of qualitative data, but yet focus and present them in a way that is readily accessible to those who are not part of academic communities" [24]. One of the intentions of this study was to be able to apply the results to the development of a model of resilience in order to be able to train and mentor women leaders on how to respond to disrespect, practice respect, and cultivate increased resilience in their work.

Purposeful sampling was employed in the study to identify women 18 years or older who had been in senior leadership positions above middle management for at least one year. Findings were obtained from semi-structured, one-on-one interviews with the 24 participants selected that lasted, on average, 60 min. All interviews took place over Zoom and were transcribed. The interviews included 18 open-ended questions. The first six questions explored who served as role models and mentors in childhood and youth and the circumstances (including their first jobs) in early life that contributed to their later success. Seven questions probed participants' work experience as a leader in their current industry. These questions examined participants' experiences of being disrespected on the job and how they responded. One question addressed participants' experience when they were forced to shift to working remotely during the pandemic. Probes covered the extent of their virtual work, how working virtually affected the range and types of disrespect they experienced, and whether and how working virtually changed the way they responded to disrespectful behavior.

These interviews allowed each participant to share incidents, stories, memories, and lessons learned from disrespectful experiences that helped shape their approach to their leadership roles. Their experiences met the test of types of disrespectful experiences such as verbal abuse, theft of intellectual property, ridicule, dethroning, racism, or sexism. Their resilience was demonstrated by the fact that these negative experiences did not cause signif-

icant setbacks, nor did they deter them from reaching their goals and succeeding as leaders. Some of the participants' relevant influences were more apparent in the circumstances and experiences they shared. For others, extra examples and probing were needed to reveal the more subtle influences. Nevertheless, all participants acknowledged a confluence of circumstances and experiences that influenced and supported their successes.

*Participant Profiles*

The 24 women in the study were previously or are currently highly successful in a variety of industries. Many of them have attained the highest positions possible in the corporate and non-profit arenas. Some of them have held critical roles for their organizations throughout the Americas, Europe, and Asia. They have all led interesting and multifaceted lives. Twenty-one (86%) combined their careers with families and three (13%) were the primary providers of the family while their partners were the caretakers of the children. Two of the participants were supported by their partners when they relocated their families overseas for career opportunities. One traveled abroad extensively, and her partner was the primary stay-at-home parent.

Fourteen (58%) of the women in the study are currently married, and all but three have children. Five (21%) are divorced, four (17%) are single, and one is a widower. The typical participant had 6–10 years of experience and an advanced degree. Seventeen (71%) women hold advanced degrees, including three with PhD degrees, two with the JD, and 12 have master's degrees. Figure 1 shows participant education levels. Eleven (46%) of the participants are in the Mid-Atlantic region of the United States, four (17%) are in the Southeast, four (17%) are on the West Coast, two (8%) are in the Northwest, and two (8.3%) are in the Northeast.

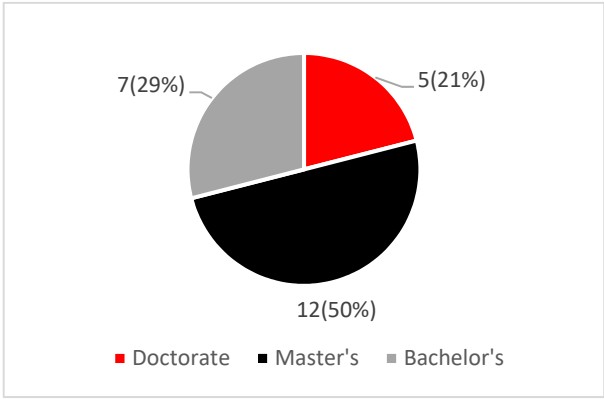

**Figure 1.** Participant Education Levels.

The women are diverse ethnically and culturally and range widely in age. Two-thirds of the women, 15 (63%), are White; five (21%) are Black: and the remaining women included two (8%) Latinx and two (8%) of Asian descent. While not planned, the race/ethnicity of the women closely reflected the current general population of women in senior leadership positions in the United States. Two (8%) women are ages 35–42, four (17%) are ages 43–50, seven (29%) are 51–58, with the balance of 11 (46%) at 59–70+. The age distribution is reasonable for a group where one-third (33%) of the women have 21+ years of senior management experience. Two (8%) have 11–14 years, seven (29%) have 6–10, and the remaining seven (29%) have 1–5 years of experience. Figure 2 presents the participant years of senior management experience.

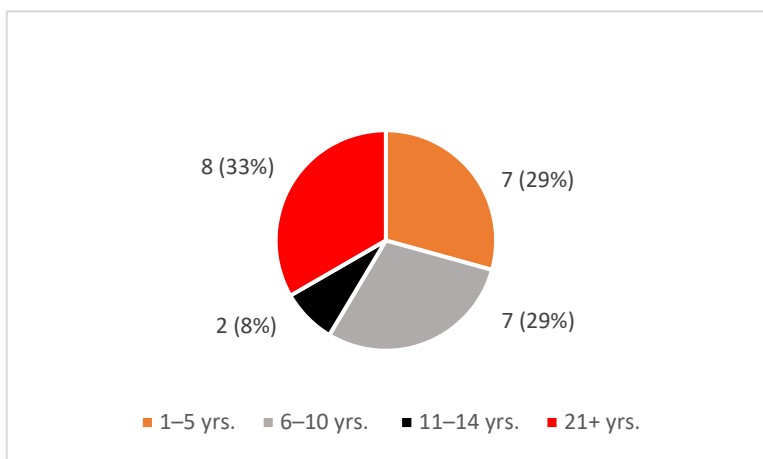

**Figure 2.** Participant Years of Senior Management Experience.

During the interviews, participants were asked to define their experience of disrespect on a continuum from low levels of disrespect (high respect) to high levels of disrespect (low respect) and give examples of each level. The interviews produced narratives containing rich descriptions of how these participants' lived experiences led to their professional success. They shared childhood experiences and described the significant people or support groups who influenced their resilience and desire to thrive. They shared their perspectives on the barriers that they experienced in the workplace and how the support (e. g., work-life balance) they created helped them to maintain their mental and physical well-being.

**3. Findings and Discussion**

Exploring the factors that impact the resilience of women leaders was accomplished by examining how ongoing experiences of disrespect in the workplace influenced their behaviors and actions over time. The participants dealt with disrespectful situations in various ways during different periods in their lives. For example, one participant began the interview with a low level of self-confidence because she had recently been unexpectedly laid off from a senior vice president position of 20+ years. Toward the end of the interview she said, "As a woman raising children, you forget who you are and how you've bounced back from many professional situations earlier in your career. This process has reminded me of who I am and what I am capable of achieving."

Findings from the interviews were designed to elicit descriptors of the participants' experiences of disrespect. The findings were identified from themes that emerged from the patterns in the data. The data were coded to indicate the participants' descriptions of their experiences in senior leadership positions in their respective workplaces.

The themes and descriptors are presented below in Table 2. Both themes and descriptors are presented in the order of their significance to the research question. The data suggested that specific descriptors within the resilience theme were more predominant than others. For example, the motivation to succeed, both extrinsic (the support systems from earlier in their lives and presently) and intrinsic (the internal fortitude that they have the tools to succeed), appeared to be the most prominent descriptors.

*3.1. Varying Definitions of Disrespect*

All participants in the study provided their own definitions of disrespect. Seven participants (29%) said that defining disrespect was difficult to articulate. That is, it is something that you know when it happens to you. One participant said it was actually harder than she thought it would be to define. Their definitions fell primarily into four main categories.

1.   Not being listened to: which could be in the form of being ignored, not being heard, unwillingness to hear what you are saying (while making eye contact)

I would state my point and they would just look at me and then continue talking. I called it "invisible woman syndrome."

2. Not being respected at the table: being excluded, diminished, overlooked, or blind-sided (agreeing with you initially and doing the opposite of what was agreed upon)

I think it is either an overt or covert demonstration of a lack of appreciation for, the gifts or qualities that another person has to contribute.

3. Not being acknowledged: purposely being ignored, being talked over, being inter-rupted, being downplayed, questioning everything you say, backstabbing, passive-aggressive type behavior

The organizations I've been a part of disrespect is something that most people feel that they are being respectful because they are being polite. But then you are left with disregard, which is not having an awareness for someone's presence, not being thoughtful about your word choice, not being curious about someone's perspective.

4. Condescension: assuming that, because you are a woman, you will take notes, order lunch, clean up after the meeting, while you are at the same level and sometimes at a higher level

Someone that violates, explicitly violates your values, your sense of well-being so that they can be seen or, in a manner that is oppressive, in a manner that is overbearing. I would say that would be disrespectful. Intentionally trying to undermine the person for their well-being, their betterment.

**Table 2.** Major Themes with Descriptors in Order of Significance to Research Question.

| Major Themes | Descriptors |
|---|---|
| Disrespect | • Participants had varying definitions<br>• Experiences with disrespect ranged widely<br>• Types of disrespect emerged<br>• Did the experience or feeling of being disrespected impact job performance negatively |
| Respect | • Definitions and types of respect emerged<br>• Experiences with respect ranged widely<br>• The experience or feeling of being respected impacted job performance positively |
| Resilience | • Participants had varying definitions–meaning of resilience<br>• Significant people or events influenced their resilience (extrinsic)<br>• Intrinsic motivators to succeed<br>• Intrinsic forces influencing success<br>• Resilience as surviving or leaving the situation |
| Disrespect in a virtual and network-enabled world | • Changes in the dynamics of feeling Level 3-5 disrespect<br>• The manner in how it is handled by the participants changed<br>• The manner in how others are disrespectful to the participants changed |

*3.2. Handling Disrespect in a Virtual and Networked-Enabled World*

The participants acknowledged that disrespect also exists in a virtual environment such as experienced during the pandemic lockdown and as would be experienced in a more flexible, post-pandemic hybrid workplace. While most felt that there was no difference regarding disrespect in a virtual and network-enabled world, some felt that they had more control over how they dealt with it there. For example, if someone disrespected them, once the meeting ended, they did not have to continue seeing the individual and could disconnect from the situation sooner. One participant of African American descent said that the feeling of being invisible was exacerbated, often requiring her to insert herself

into the meeting dialogue. Here are examples of how participants handle disrespect when working virtually and how the experience of disrespect may differ.

> I write out exactly what I feel. And then I'll go back, and I'll cross out everything that looks emotional. And then I'm left with some bullet points. And then I'll move those bullet points over, and I'll think through, what is it that I want to say? What is at the essence? What's the root of what's going on here? Then it allows me to have a conversation [about] where's the impact of what happened; I'm sure that wasn't your intent. But let's talk about it. And for some people they appreciate that; for other people, it definitely changes the relationship dynamics.

> I think that the disrespect can come in when male colleagues or male leaders do not recognize how much these individuals are balancing in addition to everything you are asking them to work in a virtual world, where it is not as easy to get it done.

> I think that there is a greater sense of safety working from home and the comforts and the safety of someone's home. But I also feel that could be a double-edged sword where someone might feel more vulnerable also.

> I think in some ways it's similar where you don't take the extra steps to hear people's perspective. I think you just see people that are better—want to make sure that we're not criticizing people [who] maybe aren't the cleverest communicators for being disrespectful, right? Everybody is not comfortable getting on Zoom calls and, you know, doing everything that we're trying to do to keep people motivated during COVID. So, I think, you know, it's a very hard place to claim somebody might be disrespecting you because they just may not be comfortable in this situation, they're in.

### 3.3. Did the Experience or Feeling of Being Disrespected Impact Job Performance Negatively?

It is telling about the resilience of the participants that none of them described negative impacts on their job performance from experiences of being disrespected. For example,

> I remember one time he asked me to do something. And I was so upset, and I went back to my desk and he walked by, and he said, "I know you're not pouting." He was like we do not pout in the workplace. He was like you're going to shake it off and get it together and we're going to walk down the hall, and so it was very hard. Yet when I left working with him, I didn't realize how much I knew. How much I had learned. And he would give me books to read, and then quiz me on it. And then he would say you know, what did we say about the seven habits? First things first. So, you know, just . . .

Some participants noted that they had a strong network outside of work, or other means of dealing with the experience of being disrespected in the workplace. One participant indicated that having a glass of wine during those tough experiences was enough to release any negative feelings or thoughts. This finding differs from the surveys previously referenced that found a connection between disrespect and job performance. The difference could be explained by the fact that participants are already successful women leaders who apparently have developed the resilience required to advance despite incidences of disrespect.

### 3.4. Definitions and Types of Respect Emerged

Participants were encouraged to give examples of the types of respect that they experienced. A content analysis of participants' definitions of respect revealed several words used consistently by them. These responses were classified into three types by frequency: (1) self-awareness ($N = 86$), (2) other-awareness ($N = 79$), and (3) allyship ($N = 54$). These are depicted in Table 3. Using pseudonyms where needed, Table 3 shows

the words that participants used and provides examples. In some examples, more than one type of respect was demonstrated.

**Table 3.** Types of Respect and Participant Examples.

| Types of Respect | Participant Examples |
| --- | --- |
| Self-Awareness–Curious, open to different viewpoints, eye contact, attentive, introspective, body language, listening | Acknowledge the fact that you are in the room or something like just acknowledging. Truly listening and engaging in the conversation, and respect doesn't mean you agree. It means you listen to the perspective. Someone that is introspective, that acknowledges when they may have said or done anything that devalued someone's time, space, energy, skill set. I'm not saying that perfectly flawless interactions, but introspection to be able to hold themselves accountable and of course, correct as necessary. |
| Other-Awareness–Recognizing my skills, talents, personality; acknowledgement, speak to me as a peer, supporting, open to different ideas, open to different viewpoints | Acknowledging the experiences that I bring, and asking me to, to support or share and sometimes lead because of what I bring to the table. I think it's truly listening, right? Truly listening and engaging in the conversation, and respect doesn't mean you agree. Right? It means you listen; you listen to the perspective. And I'm trying to think–there is a time where you're listening and engaging and acknowledging. You are the kind of leader I want to follow. You're doing a great job. Saying that in front of other leaders in our organization. I mean becoming the first female managing director in the organization. |
| Allyship–Confirmation from peers, accepting my recommendations, reinforcing, making commitments to advancing my career and doing it | There was another gentleman, who was present at the earlier meeting but had nothing to do with the incident. He approached me at the party saying that he was horrified that he didn't do the right thing by saying no we can't do that [exploit women]. He said, "I am so sorry." I said, "You didn't do it." He [replied] "Yes, I did because I didn't speak up and say that it's not okay to do." [His genuine remorse] made me feel very respected. I guess he went back to this person and told them that I was insulted. The next day I got a phone call saying, "I really did not intend to insult you." This other gentleman who had been interrupting me started speaking again. The CEO said, "Hang on a second, Tim. I want to hear what Genevieve has to say. I think she may have an interesting point." |

*3.5. Being Respected Impacted Job Performance Positively*

In response to the question about how they reacted to being respected, participants responded that they felt more committed to the organization. Because they felt safer in their workplaces, they could be more creative and productive.

> Our COO, Becky, at the time, said, "I got a great role for you. It's a VP role. It's in sales operations. I know it's not sales, but it's operations. It's something different." And she says, "It's going to be a risk for you because you don't know operations. You've done sales. You've done strategy. This is going to round you out as a general manager or COO candidate because now you're going to understand systems tools, supply chain, you know, whatever. I think you can do it."

I think for most of my career I've been in the level one and two, [high respect/low levels of disrespect] which is nice. I was put on the leadership team at my first foray into running a department at a company in the Finance Industry. It was in direct relationship to something I had done. I had redone the compensation plans and saved the company just roughly a quarter of a million dollars annually. And the president of the organization recognized that my decision and my foresight and my work had an impact that went straight to the bottom line. Shortly after that, that he made the recommendation that I be put on the executive committee, I was at 28 years old. I've been in leadership for a long time.

I was reporting to a senior VP, and I had a conversation about how we could restructure the work that we were doing so that it could probably earn higher income and he loved the idea. He called a meeting of the executive team and asked me if I "would write a proposal explaining what it would look like." And I did. When it was time to present it at the meeting he said, "Come with me because I'm not going to be able to talk about it the way that you can." So, I got invited to this executive meeting with all executives at the table and I got to pitch this idea for a new revenue stream for the organization.

And after I was finished, I remember the Chief Financial Officer saying, "I really appreciate the way your brain went into thinking about this." I felt like what I said mattered, I was valued, and my input valued. I think that's being respectful. They were respecting my years of experience, but also my mindset in terms of being able to be creative and at the same time to think about growth for the organization.

### 3.6. Development of Resilience

Participants had realistic perspectives about their industries and the lack of women in senior leadership positions in general. They had the self-esteem and self-respect that allowed them to have the self-control they needed to be effective in difficult situations. For some participants, there appeared to be a consciously deliberate, planned process consistent with some of the words they used during the interview, such as "I was very competitive" and "I have a strong desire to win."

There were major areas of influence in each woman's process in learning to be resilient when faced with a perceived lack of respect. In the participants' reflections on the development of their own resilience, they identified extrinsic influencers and intrinsic motivators that helped them retain a sense of resilience and control. They also discussed resilience as the choice to survive or leave a disrespectful situation.

### 3.7. External Influencers of Resilience

External influencers of resilience refer to the individuals who influenced them growing up and, in adult situations, those who nurtured their self-respect and the strength to confront disrespectful situations without allowing such situations to harm their professional advancement. More than half (58%) of the participants stated that the significant people who influenced them were women. These included their mothers, grandmothers, aunts, and/or women in the workplace. The remaining women (42%) indicated that the significant people in their life who influenced them were male. These included fathers, grandfathers, other male relatives, and men in the workplace.

### 3.8. Internal Forces Influencing Success

Internal forces refer to motivational forces that are inherent to the self or the task. For the participants whose motivators were less conscious, the analysis suggested that they were primarily related to people they knew and circumstances that occurred early in their lives. These associations served as an unseen force that kept the participants motivated to persist in pursuit of their goals when faced with disrespect.

These early influences in their personal lives informed their perception of gender roles and gender expectations. For example, one participant's father was a strong influence

because he essentially believed that women should not be molded to be a certain type, even though she grew up in a country where women were expected to play the role of homemaker. Another participant grew up in a small town in the South, and her grandfather taught her how to drive when she was six years old and not tall enough to see over the steering wheel. That experience became a reminder later in life when she found herself in difficult circumstances and was unable to see how a successful outcome was possible. For example, she married at a young age, divorced, and was left to raise a child as a single parent by the time she was 18 years old.

One participant did not remember consciously setting goals, yet she recalled two circumstances that stood out in her memory as an adult. The first was when she was a young girl. The women in her family either worked as grocery store clerks or as housekeepers cleaning other people's homes, except for one aunt who made a big impression on her. Growing up in the Midwest she remembers her aunt's company flying her to the East Coast for meetings during a time when few women worked in organizations beyond clerical workers. She had opportunities to visit her aunt and admired the lifestyle that she attained professionally.

The second circumstance was learning how to challenge herself and take risks, a skill that proved valuable as an adult. Because both of her parents worked, she and her brother were without adult supervision when they returned home from school, often referred to as latchkey kids. She was the oldest and therefore in charge. Her brother and his friends did not allow her to play with them. The boys would mainly play softball and girls could not play. Jeanie said this was her earliest remembrance of disrespect. She found ways to entertain and challenge herself by taking risks without any awareness of putting herself in danger. These experiences helped her develop characteristics and take risks that helped her succeed professionally.

### 3.9. Resilience as Surviving or Leaving the Situation

Some of the participants were exposed to resilient women, giving them permission to expand their visions for themselves and pursue their professional goals and aspirations. One participant shared that her mother is an activist, and she still gets the opportunity to hear her speak about inequities and injustice and continue to do something about it. With admiration she stated, "She's not a person that gives lip service." This participant went on to say that her mother gave her the tools and strategies to deal with the world that in most cases "is not built for people [African American] like me."

Participants highlighted that they had the ability to recognize that they had choices in every situation and had the option to stay or leave.

I remember thinking, well I don't know what to do about that. I'm not leaving my job, so you just get over it. Well, that's a sassier reply than I would have had then, [even though that is what I did].

But there is always that choice you have to make when you are confronted with certain situations where whatever you try will not work with a person, and it is enabling them to continue to be abusive in some way or disrespectful in some way, and then you make a different choice.

I just really felt like if I cannot be in my own truth in this role and in this organization, then this is not the place for me. And I think being solid in that belief is what helped me recover from that.

When I took the job, I had a boss that could not have been better for me. He was hands off. He gave me a project and let me go. I knew instinctively that once his successor took over, it was not going to work. His successor was completely the opposite. He was micromanaging. He was misogynistic. He was very much a narcissist and a tyrant. An interesting example was one of the first meetings I had with him, he was prepping me to go into a meeting. Which is great guidance from a CEO. But at one point he looked at me and said now here's what I want you to

say. Been managing to talk since I was about two. Haven't really needed anyone to instruct me. And I left. Actually, it was at that instant I made my decision to leave.

### *3.10. Personal Resilience as a Skill Set*

The resilient behaviors that the participants exhibited to attain and retain their successes are skills that can be developed through coaching, mentoring, and training. The study revealed six mindsets and strategies underlying their successes as illustrated in Table 4 below.

**Table 4.** Resilience Skills: Mindsets and Strategies.

| Mindsets and Strategies | Resilient Behavior |
|---|---|
| Presence: In myself, and I nurture it in others | -Self-Awareness-I am mindful of the importance of communicating and interacting respectfully with others.<br>-Other-Awareness-I pay attention to the cues from others to let me know if they feel respected or disrespected.<br>-Allyship-I am willing to take a stand on behalf of others when I see that they are being disrespected regardless of whether or not they are present. |
| Commanding Respect | -I no longer see myself as powerless.<br>-I am giving myself permission to step into my power. |
| Follow the Secret Sauce Recipe | Internalize and exhibit the resilient characteristics I identify with:<br>-Perseverance and faith<br>-Act like a Level 1 leader<br>-Courage<br>-Find common ground<br>-Don't take a lot of crap |
| Gratitude | -I am grateful to all the positive role models and influences in my life.<br>-I tap into my deepest emotional roots. |
| Self Before Service | -I know I always have choices, even if they are difficult ones. |
| Respect and Protect My Well-Being | -I pay attention to my mind, body and emotional cues and take action when I feel out of balance.<br>-I have practices that help me maintain my purpose and connection to that which is greater than myself. |

## 4. Summary and Implications

The goal of this study was to explore the factors that impact the resilience of successful women leaders who experience disrespect in the workplace. Factors that emerged included early developmental influences, circumstances they faced in their youth and young adulthood, and experiences that shaped how they responded as adults to disrespect or a lack of respect from their organization's stakeholders (e.g., board members, leaders, peers, subordinates, and clients). The study showed that there are five factors that help women leaders develop resilience, including experiences, beliefs, values, people, and events shaping their lives. This study revealed the complexity and variation in the experiences, beliefs, values, people, and events shaping the lives of women in senior leadership positions who are resilient when faced with disrespect in the workplace, including in a virtual environment. Despite these complexities, the study helped to formulate a clearer understanding of how disrespect and respect in the workplace are experienced and whether they impact performance. The women leaders in the study indicated that they did not allow disrespect to impede their performance and that respect did serve to improve their performance.

Although the study revealed that one's upbringing, caretakers, and the influence of certain individuals in one's life contribute to preparing women leaders to being resilient, the study implies that women leaders can still learn certain strategies and work to master particular mindsets to help them become more resilient in the face of disrespect in

the workplace. The implication of these findings is that resilience can be learned to a large extent.

In addition, the pandemic has led to the withdrawal of millions of women from the workplace and their reinsertion into the workplace is moving slowly [25]. There is recognition that changes will need to be made in the post-pandemic workplace for women to be more engaged and satisfied with their work [26]. The implications of this study for the post-pandemic workplace includes the recognition of the importance of respect and the need for organizations to ensure a respectful work environment.

## 4.1. Study Significance and Limitations

The study adds to the growing body of literature regarding the characteristics and importance of resilience to the success of women leaders in senior leadership positions. The study also reinforced the importance of respect in the workplace and the potentially negative impact of disrespect on job engagement and satisfaction.

The study was limited in that study participants were already senior women leadership who had successfully exercised resilience against disrespect in the workplace. In addition, the study sample was relatively small and being a purposeful sample, the findings may be limited to the selected group of participants. Because the study was conducted during the COVID-19 pandemic and all interviews were conducted virtually, narratives may have been somewhat different than they would have been if elaborated in face-to-face interview situations where the relationship between researcher and participant could be more personal.

## 4.2. Recommendations for Future Research

Future research should be carried out on the experience, attributes, and importance of resilience among professionals and workers at all levels in order to understand all the nuances of how one develops and exercises resilience in the face of workplace disrespect. Further, research on the development and impact of self-respect on resilience and success in the workplace should be studied more. Research should include a focus on the role self-respect plays in the organizational context and whether self-respect can be cultivated in adults. Researchers can clarify the difference between self-esteem and self-respect. Self-respecting individuals are motivated by a fundamental belief that, despite their circumstances, they and others are equally unique contributors to the greater whole [27]. This has implications for understanding, developing, and enhancing resilience, independence, and strength in individuals from marginalized groups in the workplace and should form the basis of future research [28].

## 4.3. A Suggested Potential Model of Disrespect, Respect, and Resilience

Based on the findings of the study combined with findings in the literature, a more comprehensive model of disrespect, respect, and resilience is proposed to form the basis of training, mentoring, and coaching. This model is presented in Figure 3 below:

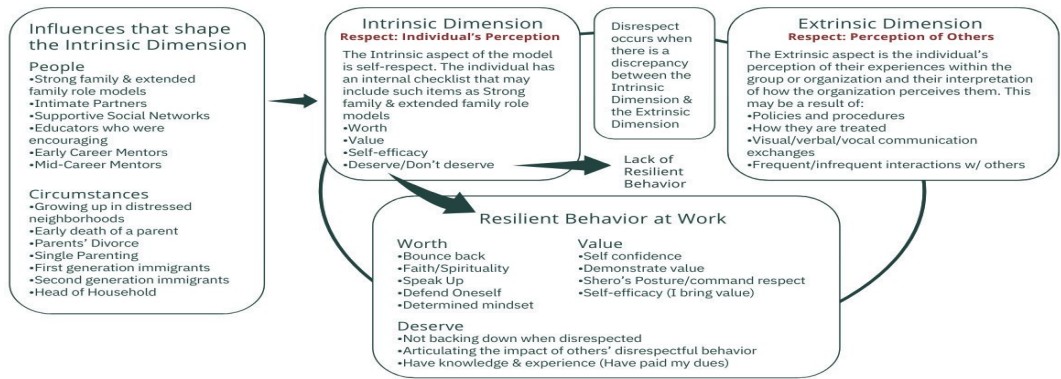

**Figure 3.** Disrespect/Respect/Resilience Model.

The intrinsic dimensions are developed by the influences that the individual experienced in various aspects of their lives and may include such items as family and extended family role models.

When those influences are strong and positive, the individual believes that she deserves respect and displays resilient behavior in the workplace because she has a sense of self-worth. She can bounce back quickly from adversity, speak-up, defend herself, and maintain a determined mindset to succeed. She sees her own value. She is self-confident, recognizes and demonstrates her value to others in the workplace, and commands respect by her verbal and nonverbal communication. She feels that she deserves to be respected, she does not back down when disrespected, and she is willing to assertively articulate the impact of others' disrespectful behavior.

When those influences are strong and negative, the individual believes that they are unable to display resilient behavior. She operates instead from a place of not deserving respect and relies on the extrinsic dimensions, the perception of others, to determine her self-worth and lacks resilient behavior

This Respect/Disrespect/Resilience model can help identify areas for improvement or strengths as it relates to feelings of being respected from the perspective of women leaders. The model can be used to develop leadership programs for high-potential women leaders from diverse populations to increase their level of self-respect and self-esteem. The model can be used to describe the dimensions of self-worth, or how women leaders can overcome lifelong experiences of disrespect by strengthening and holding on to their self-worth.

## 5. Interview Protocol

Personal Resilience in Women Leaders who Experience a Lack of Respect in the Workplace

(a) Research Question

What factors impact and account for the resilience of women leaders that experience disrespect in the workplace?

(b) Introductions

Hi (research participant). How are you today? Thanks for agreeing to talk with me and to participate in my study. This conversation is going to take about an hour. I know how valuable your time is. I appreciate the time you are taking out of your busy schedule to talk with me today about your experiences in dealing with situations where you felt a lack of respect (or disrespect) from the leaders, your managers, peers, or direct reports at work. I'll do my best to make good use of your time.

I am looking for your personal perspectives on your experiences in your leadership position. I encourage you to feel free to share with me your personal thoughts, feelings, and beliefs.

All your answers will be held in the strictest of confidence. I am committed to maintaining your individual confidentiality and will address how I will do that shortly.

Let's start by introducing ourselves. I'll go first. I have a personal and professional interest in women's leadership development. I received the doctorate in Human Development from Fielding Graduate University in April 2022. I have a master's in Human Development and a master's Organizational Behavior.

Professionally, I have worked in higher education, and for non-profit organizations and corporations in such areas as employee development, social equity, organizational development, and culture change. I've held positions as Vice President of Finance and Administration, Dean of Intercultural Development, and Vice President of Organizational Development and Culture Change prior to launching my company, The Folke Institute for Transformative Learning in 2004.

Now it's your turn. Please tell me a little about yourself. (Research participants introduce themselves.)

Thank you.

(c) The Research Study

Let's talk a little bit about the research study you are participating in. Through your participation, you are helping to expand the research regarding the factors that may support or hinder women from advancing to senior level positions within their respective organizations, such as resilience. There is a great deal of emphasis on the extrinsic factors that hinder women from advancing into leadership positions in the workplace, but I'm interested in your experience.

(d) Interview Agenda

Here's how I've structured our meeting together. First, we'll take about five minutes to get a few procedural things out of the way. Then, we'll shift into the questions about your experiences of lack of respect (disrespect) and how you have dealt with barriers to your progressing within the agency. That will take up to 45 minutes. When we're done with the questions, I'll let you know what will happen next and how you will be able to stay in touch, to the extent that you want to, with my research as it progresses. You can also share anything with me that did not come up in the interview about your experiences.

(e) Informed Consent and Procedural Information

Let's get the procedural things out of the way.

- First, I need to review the Informed Consent form with you that you have signed and returned to me. I want you to understand that everything you share with me is completely confidential.
- I will be recording our discussion in order to capture your full response and will also take some handwritten notes as a back-up. I want you to know that you have the option to pass on any question for any reason and you can end your participation in the interview at any time and ask that your information not be included in the study.
- The digital of this discussion will be stored in a protected format and then transcribed. I will send you a copy of the transcript from this interview and you will then have an opportunity to review the transcript and make any corrections or clarifications. Within the transcripts you will be identified by a different name, a pseudonym, to protect your identity. I will ask you to select a pseudonym at the end of our discussion.
- Do you have any questions about what I've just explained or any other aspects of the study? (Answer questions get signatures and give copy to participant). Well, that's it for the procedural stuff. Let's get into the interview itself. I'm going to start recording now. Ok?

    Before we get into talking about your experience in your current role, I'd just like to learn a little bit more about you.

1. Was there a person or circumstance in your early life that helped you to succeed professionally?
2. How would you describe yourself in terms of setting goals and achieving them growing up?
3. What there a person or circumstance when you were just starting out that helped professionally?
4. Tell be about your first job?
5. Can you think of a time when you set a goal and did not achieve it because you were rejected due to no fault of your own? If so, how did that make you feel? How did it impact your confidence? Self-esteem? Etc.
6. What contributed to your decision to pursue your current profession? (Pre-retirement profession?)

    Now let's talk about your experience as a professional in your current industry.

7. How long have you worked for your current organization?
8. How long have you been in your current or previous leadership position?

    Thinking of respect on a continuum from high to low levels of respect (Level 1= I experience this person or situation as demonstrating no respect. Level 3 = some respect; Level 5 is demonstrating high levels of respect:

9.  How do you define a lack of respect or the absence of respect? What behaviors do you equate with a Level 1 Respect? What about a Level 5?
10. Describe a time when you felt you were treated at a Level 1 at work.
11. Describe a time when you felt you were treated at a Level 5.
12. When faced with high levels of disrespect, what helps you be resilient?
13. Tell me about a time when you felt respected at a Levels 1–3.
14. Give an example of a time that stands out for you when you felt Levels 4–5 disrespect in front of your peers, direct reports? Senior management?
15. Have you ever taken any legal actions in response to your feelings of Levels 4–5 disrespect?
16. In a virtual and network-enabled world, work gets done in a variety of places, often outside of the workplace.

    a.  What percentage of your work gets done in the workplace office?
    b.  What percentage of your work gets done out of the office (including working from home/virtual versus face-to-face meetings in other places with people?
    c.  Does the variety of workplaces change the dynamics of feeling Levels 3–5 respect by your direct reports, peers, manager? Senior managers?
    d.  How does the virtual environment change how you handle disrespect? Have you found anything that really works well?

17. Tell me about a time that you are most proud of in your ability to successfully respond to disrespect in the last 18 months.
18. Additional thoughts/comments?

**Funding:** This research received no external funding.

**Institutional Review Board Statement:** The study was conducted in accordance with the Declaration of Helsinki and approved by the Institutional Review Board (or Ethics Committee) of Fielding Graduate University (protocol code 19-0101 and date of approval 15 January 2019).

**Informed Consent Statement:** Informed consent was obtained from all subjects involved in the study.

**Data Availability Statement:** Research data are available at Spell-Hansson, C. (2022). *Unbossed and unbroken: Personal resilience in women leaders who experience a lack of respect in the workplace* (Publication No. 29165095) [Doctoral dissertation, Fielding Graduate University]. ProQuest Dissertations and Theses Global. https://www.proquest.com/docview/2659621419/DA6C27951C24966PQ/1?accountid=33310, (accessed on 9 September 2022).

**Conflicts of Interest:** The author declares no conflict of interest.

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
