# Peer review of "Key Factors That Contribute to the Development of Resilience in Successful Women Leaders Who Experience Disrespect and the Importance of Respect in the Post-Pandemic Workplace"

_merits, doi:10.3390/merits3010009_

Round 1

Reviewer 1 Report

Dear Author:

While the premise of the article is excellent, there are several serious flaws.  

First, it is stated in the abstract "The study examined study participant experiences of disrespect and respect in the workplace prior to and during the pandemic, including how their roles changed when their organizations shifted to working virtually. The study also probed the intrinsic and extrinsic factors which contributed to the participants’ resilience when confronted with disrespect from their leaders, peers, or direct reports [1]."

In this wording, there is a citation [1]. It appears that citation [1] is this study. You don't cite the current article in the list of references.

These are all competing statements with different foci. There must be specific research questions that are being asked. 

It is included in those sentences "including how their roles changed when their organizations shifted to working virtually" and yet this analysis is not performed with the data. It's not the only analysis that was promised, but not performed, with this study.

Second, The introduction is missing a significant amount of information. You have not described the four clusters of workplace mistreatment, the fundamentals of workplace mistreatment, women at work, or the outcomes of workplace mistreatment. Articles such as https://www.researchgate.net/publication/359945255_Workplace_mistreatment_for_US_women_best_practices_for_counselors can help you understand these concepts.

Third, it is stated that the analysis of "qualitative methodologies of narrative inquiry, thematic analysis, and content analysis" was used. This is impossible. Narrative inquiry, thematic analysis, and content analysis are all different analyses. As such, what is listed as the analysis is not appropriate. A couple of articles that can teach you how to do content analysis can be found at https://www.sciencedirect.com/science/article/pii/S2211419X17300423 and https://www.sciencedirect.com/science/article/pii/S2352900816000029

Fourth, all of the questions asked of participants must be listed in the article verbatim.

The rest of the article cannot be used because of the flaws in the four areas listed above. If the four areas are addressed above, you will have to completely start over from line 196 onward.

Reviewer 2 Report

Overall comments:

This was a really interesting study that I enjoyed reading. The topic of study is important, especially due to the changing nature of work and how women were disproportionately affected by the pandemic. Overall, throughout your manuscript, you need to incorporate more theory. Respect and resilience are two concepts that have been well-studied from a variety of perspectives and are rooted in respect and resilience theory. Clearly understanding and conveying the theoretical underpinnings of respect and resilience will make your paper stronger and will help you make a stronger claim for how your model (amazing!) helps to advance these theories.

Introduction

·      Typically, especially in qualitative research, papers don’t start with the goal of the study. Instead, include relevant literature and/or describe the current situation that clearly justifies the importance of conducting original research to further explore the topic. Then, follow with the goal/purpose of the study. For example, why the focus on respect/disrespect and resilience? Cite previous studies that provide logical reasoning for this focus.

·      Need supporting citations for the disruption of work and feelings of discontent (lines 36-39)

Respect and disrespect

·      Need more citations supporting the argument about women’s’ experience with disrespect in the workplace. You repeatedly cite the McKinsey report, but more research exists to offer a more well-rounded perspective.

·      I suggest looking into the literature on workplace respect theory to define respect and disrespect in this section. See:

o   Rogers, K. M, Corley, K.G, & Ashforth, B.E. (2017). Seeing more than orange: Organizational respect and positive identity transformation in a prison context. Administrative Science Quarterly, 62(2), 219-269.

o   LaGree, D., Houston, B., Duffy, M., & Shin, H. (2021). The effect of respect: Respectful communication at work drives resiliency, engagement, and job satisfaction among early career employees. International Journal of Business Communication, 23294884211016529.

Resilience

·      No comments here. This was a great section.

Materials and Methods

·      Where are the research questions? Based on the literature, and the topic you’re studying, what question or questions do you intend to answer? These should be included after the literature review.

·      You need a clear justification for why qualitative research and the specific methods you outlined are appropriate to explore the research questions. Include citations from articles on qualitative methods here.

·      Participant profiles and description of the sample was very detailed. Good work here.

Results

·      Need an introductory sentence or two here explaining the general themes you uncovered.

·      Incorporating some direct quotes from participants in you’re the “varying definitions of respect” section would help readers have a better understanding of your themes. And, this is a key benefit of qualitative research.

·      Enjoyed the content in Table 3. Good work, and a great way to summarize extensive qualitative data.

·      Figure 3 is amazing—the model builds and extends theories of workplace respect and resilience

Discussion, Conclusion and Implications

·      You do a fine job summarizing your results. What’s missing is a clear explanation for how this extends our knowledge and understanding of how respect and resilience works in the workplace, with a focus on how your findings and model also extend/advance respect and resilience theory.

Reviewer 3 Report

This paper explores the issues of respectful and disrespectful behavior against successful women in corporate leadership. The author(s) conduct interviews with such women about their experiences and responses. These insights are tied together into a model (Figure 3). The authors also make interesting policy recommendations for how to strengthen gender minorities in the workplace.

The topic of the paper is clearly important and I learned a lot from reading it. The empirical work was clearly described and I understood the theoretical concepts used as a basis for the data collection. Overall, the paper gives useful policy advice and contains a lot of interesting and valuable theoretical discussion of the empirical results.

A point of confusion to me was the theoretical model that appears at the end of the paper (Figure 3). I was not sure what the purpose of this model was or how it related to the empirical results. Looking at the figure, I get the sense that it wants to explain why disrespect occurs? But that is not what the empirics is about; to me, the empirics are more interested in explaining why resilience occurs. The Figure’s circular pattern of factors was also unclear to me and I was not able to link the “boxes” to specific empirical sections that I had read. For example, there is an empirical section on “extrinsic factors for resilience”, but the figure has extrinsic factors for respect; and I was not sure why disrespect emerges as a combination of the “intrinsic” and “extrinsic” dimensions. Do you mean, for example, that if a woman has higher self-worth, the same behavior against her is less disrespectful?

Clarifying the purpose of the model and how it relates to the empirics in the paper would be useful, I think. The current text says that the “model emerged” based on this paper’s findings combined with other literature, which was a bit too vague for me to understand in terms of which methodological steps were taken to derive it.

For the paper’s introduction, a possible re-write could consist of explaining the paper’s contribution(s) related to the previous literature, especially to other interview studies with similar research questions, if such studies exist.

Round 2

Reviewer 1 Report

Dear Author:

This manuscript is coming along. However, there are still major issues that must be addressed.

There are significant grammar and punctuation issues throughout the manuscript that must be fixed.

It is confusing that the first citation in the manuscript is a 7, it must start with a 1. All the citations must be renumbered.

In line 201 - I do not understand what autonomous respect is. It must be defined in a way that can be understood.

In lines 212-216, this cannot be one long sentence. Must break apart into multiple sentences.

The Method section is missing significant amounts of information. I have attached a dissertation that was done using Narrative Analysis. Use Chapter 3 of the dissertation as a guide to redoing this section.

The figures in the document are not appropriately labeled and do not appear in the correct location in the document.

At line 361 there is an * but there is no explanation for what the * means.

The tables at 361 and 465 are not labeled.

What is written in line 593 does not make sense. You did not just look at intrinsic factors. 

What is written in line 600 about "five factors" comes as a shock as it does not at all align with what is written previously in the Results section.

The model presented in line 613 does not make sense. It comes out of nowhere and does not align with what was written previously in the Results section.

The entire results section is incomprehensible and must be completely rewritten. 

What is written in lines 616-631 does not make sense. It comes out of nowhere. This was not studied. It all must be cited, but I do not understand why any of this is in here. 

Where does the table in line 645 come from? If it is from this study, then it must be placed in the Results section. 

I have attached a dissertation that was done using Narrative Analysis. Use Chapter 4 of the dissertation as a guide to redoing the Results section.

I don't understand the Discussion section at all. This section currently appears to be about "Future Research." There are no discussions of the limitations of the study or a real discussion section.  The discussion section must be rewritten.

The Implications are not thorough or deep with specifics about what should be done. This section must be redone.

I have attached a dissertation that was done using Narrative Analysis. Use Chapter 5 of the dissertation as a guide to redoing the discussion section through the end of the manuscript.

Author Response

Thank you for your excellent comments. I have attached how I have dealt with them.

Reviewer 2 Report

I want to commend the authors on their significant revisions of this manuscript. They addressed all of my concerns and the paper is now theoretically sound and presents a strong argument, through relevant literature, for the focus on resilience and disrespect among women leaders. The model and tables are impressive. Great work! I see no further revisions needed, except one final spell/grammatical check. 

Author Response

Thank you so very much for your excellent comments on my first submission. I learned a lot and appreciate the fact that you now think my manuscript is ok. I am assuming you think it is now publishable.